# Sufficiency and consistency of antenatal care services in Ethiopia: Evidence from Mini Ethiopian demographic and health survey

Dessalegn Tamiru[1]*, Mohammed Areb[2], Gemechis Gunja[3], Million Abera[4], Desta Asefa[4], Gurmesa Tura Debelew[1], Negalign Berhanu[1], Yibeltal Siraneh[1], Fira Abamecha[1], Biru Abdissa Mizana[4,5]

1 Faculty of Public Health, Institute of Health, Jimma University, Jimma, Ethiopia, 2 College of Medicine and Health Science, Haramaya University, Harar, Ethiopia, 3 Faculty of Health Sciences, Institute of Health, Jimma University, Jimma, Ethiopia, 4 Jimma University Medical Center, Jimma University, Jimma, Ethiopia, 5 Faculty of Medicine and Life Sciences, Hasselt University, Hasselt, Belgium

* dessalegn97@gmail.com

## Abstract

### Background

Antenatal care plays a critical role in promoting the health of mothers and infants. In Sub-Saharan Africa, about 76% of women accessing at least one visit. However, rural areas in Ethiopia face more significant challenges, including traditional beliefs, geographical distance, and the need for more awareness. Despite advancements, ensuring quality ANC for all women necessitates targeted interventions. There are needs of evidence regarding adequacy and consistency ANC services in different regions of Ethiopia. Therefore, study aimed to evaluate and compare the adequacy and consistency of ANC services across various regions in Ethiopia.

### Methods

Data for the 2019 Mini Ethiopia Demographic and Health Survey (Mini EDHS) were collected using a cross-sectional study design. The survey included 2,918 women aged 15–49 from various regions of Ethiopia. Data analysis was carried out using STATA version 14. Descriptive statistics, such as frequencies and percentages, were used to present the background characteristics of the participants. To assess the significance of variations and associated factors, a logistic regression model was applied. Variables with a p-value of 0.25 or less in the bivariate analysis were included in the multivariable analysis. A significance level of $p < 0.05$ and a 95% confidence interval were used to determine the significance of the variables.

### Results

The study surveyed 2,918 women aged 15–49, with 70.7% receiving antenatal care (ANC) from a skilled provider. Among them, 56.8% had four or more visits, and 43%

**Data availability statement:** Data cannot be shared publicly because it was taken from the Ethiopian Demographic and Health Statistics Center. The data is available through the Ethiopian Demographic and Health Statistics Center's Institutional Data Access for researchers who meet the criteria for access to confidential data. Any researcher can obtain this data by submitting an official request through the following link: 'https://www.dhsprogram.com/data/dataset_admin/login_main.cfm'.

**Funding:** The author(s) received no specific funding for this work.

**Competing interests:** The authors have declared that no competing interests exist.

started care in the first trimester. Overall, 35.8% received all routine ANC components, and 14.7% received adequate ANC services. ANC adequacy varied by region, with the highest in Addis Ababa (39.1%) and lowest in SNNPR (6.2%). About 72% of women received consistent ANC, with Addis Ababa showing the highest consistency (54.2%). Logistic regression analysis found that education status, region, wealth status, and literacy were significant factors influencing ANC adequacy. Regional disparities showed lower ANC adequacy in Oromia, Benishangul, SNNPR, and Gambela compared to Tigray. Wealthier women and those with secondary or higher education were more likely to receive consistent ANC.

## Conclusion

The study underscored the significant impact of regional disparities, financial constraints, education levels, and geographic location on the unequal access to and quality of antenatal care (ANC) services in Ethiopia. Promoting ANC utilization, especially among women with lower education and economic disadvantages, was identified as a key priority. The study recommends targeted interventions to address service gaps for less-educated women and highlights the need for focused efforts to reduce regional disparities in both the adequacy and consistency of ANC services.

## Introduction

Antenatal care is crucial for the health of pregnant women and their babies, occurring during pregnancy. It includes health promotion, screening, diagnosis, and disease prevention. Evidence-based ANC practices have the potential to save lives [1]. Regular ANC visits facilitate early detection and management of pregnancy complications, reducing the risk of maternal and infant deaths. ANC also improves access to skilled birth attendants, ensuring critical assistance during childbirth and promoting positive birth experiences. However, in Africa, ANC services encounter various challenges such as limited healthcare access, inadequate resources, and teen pregnancies [2]. Despite these obstacles, there have been improvements, with an average of 76% of pregnant women in Sub-Saharan Africa receiving at least one ANC visit. New initiatives, like the WHO's recommendation for a minimum of eight contacts throughout pregnancy, are being implemented. Innovative methods such as mobile clinics and community health workers are being explored, requiring collaboration across multiple sectors [3].

While antenatal care (ANC) coverage has improved in Ethiopia, challenges persist, particularly in rural areas where access is limited. Traditional beliefs and practices, as well as early childbearing, contribute to insufficient ANC services [4–7]. Other obstacles, such as distance, lack of skilled providers, and limited awareness, further hinder access to adequate ANC. Targeted interventions are needed to ensure that all women receive sufficient and complete ANC to maintain good health [8,9].

ANC coverage varies by country, and in Africa, many pregnant women do not meet the World Health Organization's (WHO) recommendation of four or more ANC visits

[3]. Numerous reviews summarize the adequacy of ANC in Ethiopia and discuss its consistency, as well as the variables affecting ANC utilization [1,10]. The WHO updated minimum suggested number of ANC visits in 2016 from four to eight contacts, based on new research indicating that increased interactions between pregnant women and healthcare professionals can lead to fewer maternal deaths and improved pregnancy outcomes and neonatal health [1,6,10,11].

Given the lack of clear evidence regarding regional variations in the adequacy and consistency of ANC service, this study aimed to assess the regional disparity and associated factors using data from the Mini Ethiopian demographic and health survey (Mini-EDHS) 2019.

## Methods

### Data source, study setting, and design

The study was conducted in Ethiopia, the largest country in the Horn of Africa and the second most populous country in Africa. The study utilized data from the 2019 Ethiopia Mini Demographic and Health Survey as its primary data source. The 2019 EMDHS was a cross-sectional survey carried out in nine regional states and two city administrations of Ethiopia from March 21 to June 28, 2019. This study was conducted from January 01 to February 29, 2024.

### Study population

The study population consisted of women who had at least one antenatal care visit during their most recent birth within the five years preceding the 2019 Ethiopian Mini-Demographic and Health Survey.

### Sampling technique

Every region was divided into urban and rural areas according to the 2019 EMDHS, resulting in 21 sampling strata. We selected 305 enumeration areas (EAs) using a probability percentage, with 212 rural and 93 urban locations among them. For effective comparison and proportionate allocation, 25 EAs were chosen from each of the eight regions. Additionally, 35 EAs were selected from the three larger regions (Amhara, Oromia, Southern Nations, Nationalities, and Peoples' Region). The households in the selected EAs were included in the sampling frame list created between January and April 2019, forming the 2019 EMDHS cluster. Using a specified equal chance probability systematic selection procedure, thirty households were selected from each cluster. A total of 2918 study participants met the requirements for inclusion in this study.

### Data collection procedure and techniques

The data collection tool used for the 2019 EMDHS was questionnaires, and the data collectors were experienced in conducting household surveys. The data were gathered through interviews with women aged 15–49 years, who were long-term residents of the selected households. Essential data for this study were extracted from the 2019 EMDHS data, with a focus on variables related to the adequacy and consistency of antenatal care services across all regions.

### Study variables

The study aimed to examine regional differences in the adequacy and consistency of antenatal care services in Ethiopia. The adequacy and consistency of antenatal care services, as determined by elements including the frequency and timing of ANC visits, the supply of necessary services, and compliance with advised protocols, were the outcome variables of interest. Residence, household wealth index, geographic region, and educational attainment, which were divided into four categories: no education, primary education, secondary education, and higher education, were among the exploratory factors. The study sought to discover potential regional differences in Ethiopia and to obtain insights into the factors impacting the provision of ANC services by analysing these variables.

## Operational definition

**Timing of first antenatal care visit (early initiation of ANC):** It is an indicator assessing the proportion of pregnant women who initiate ANC within the within three months of gestational age [1].

**Consistency of ANC visit:** Measured if the pregnant mother received services during antenatal care visits such as, identifying pre-existing health conditions, early detection of complications during pregnancy, health promotion and disease prevention, birth preparedness, and complication planning [12,13].

**Sufficient antenatal care:** Sufficient antenatal care involves the recommended minimum of four timely visits, providing comprehensive services such as screenings, monitoring, and health counseling to ensure the well-being of both mother and baby [1,12,13].

**Adequacy of antenatal care:** The adequacy of antenatal care is measured using three indicators: early initiation of antenatal care (the first antenatal care visit made during the first trimester or three months), attending enough visits (at least four and above visits), and getting services at least once during the pregnancy care) [12,13].

**Wealth index:** The wealth index weights were derived using principal component analysis (PCA) based on data collected from household assets, utilities, and other variables from the Ethiopian Demographic and Health Survey. PCA was used to create latent factors that represent wealth, with the first factor serving as the household wealth score. These scores were then divided into five quintiles: lowest, second, middle, fourth, and highest [9].

## Data quality management and analysis

The study was conducted a thorough examination of the data from the 2019 Ethiopian Demographic and Health Survey (EDHS) for errors, completeness, and consistency with the 2019 EDHS report. The analysis was performed using STATA Version 14. Initially, we identified all the explanatory variables that potentially influenced the dependent variable. The data underwent cleaning, merging, and re-coding to derive relevant variables for analysis.

The essential background characteristics of the study participants were presented using frequencies and percentages. Results were communicated through tables, figures, and text. A logistic regression model was employed to comprehend the significance of variation and associated factors. Variables with a p-value of 0.25 in bivariate analysis were included for multivariable analysis. A significant level of p-value 0.05 with a 95% confidence interval was used to determine the importance of the variables. Overall, the study process ensured the accuracy and reliability of the data analysis, providing valuable insights into regional differences in antenatal care services in Ethiopia.

## Ethical consideration

We obtained a letter of ethics approval from the DHS. It allows us to access the EDHS-2019 data set from (https://dhspro-gram.com/Data/). We followed the data-sharing policy and guidelines. We kept the data private. We used it only for this study. You can read the entire report on the ethical issue in the EDHS-2019 [8].

## Results

### Socio-demographic characteristics of the participants

The study included women aged 15–49 years who had live birth in the last year (n = 2918). Nine hundred forty-eight (32.5%) of the women who participated in the study were between 25 and 29 years. More than three-fourth (69.5%) of the women were rural by residence. One thousand two hundred and sixty-nine (43.5%) of respondents had no formal education. Similarly, 1167 (40%) had four or more young children. In the household wealth index, 904 (31%) participants were in the richest wealth quantile "Table 1."

**Table 1. Socio-demographic variables of women of reproductive age in Ethiopia, 2019E_DHS (n = 2918).**

| Variables | Frequency | Percent |
|---|---|---|
| Age group (years) | | |
| 15-19 | 169 | 5.79 |
| 20-24 | 611 | 20.94 |
| 25-29 | 948 | 32.51 |
| 30-34 | 600 | 20.56 |
| 35-39 | 392 | 13.43 |
| 40-44 | 153 | 5.24 |
| 45-49 | 45 | 1.54 |
| Region | | |
| Tigray | 322 | 11.03 |
| Afar | 230 | 7.88 |
| Amhara | 336 | 11.51 |
| Oromia | 346 | 11.9 |
| Somali | 86 | 2.95 |
| Benishangul | 299 | 10.25 |
| SNNPR** | 326 | 11.17 |
| Gambela | 268 | 9.18 |
| Harari | 247 | 8.46 |
| Addis Ababa | 227 | 7.78 |
| Dire Dawa | 231 | 7.92 |
| Place of residence | | |
| Urban | 889 | 30.47 |
| Rural | 2029 | 69.53 |
| Religion | | |
| Orthodox | 1089 | 37.3 |
| Protestant | 585 | 20.1 |
| Muslim | 1202 | 41.2 |
| Others* | 42 | 1.45 |
| Educational level | | |
| No education | 1269 | 43.5 |
| Primary | 1066 | 36.53 |
| Secondary | 356 | 12.2 |
| Higher | 227 | 7.77 |
| Marital status | | |
| Never in union | 19 | 0.7 |
| Currently in union/living with a man | 2700 | 92.5 |
| Formerly in a union/living with a man | 199 | 6.8 |
| Number of children ever born | | |
| One | 705 | 24.2 |
| Two | 610 | 20.9 |
| Three | 436 | 14.9 |
| Four or more | 1167 | 40 |
| Household wealth index | | |
| Poorest | 594 | 20.36 |
| Poorer | 503 | 17.24 |
| Middle | 457 | 15.66 |

*(Continued)*

**Table 1.** (Continued)

| Variables | Frequency | Percent |
|---|---|---|
| Richer | 460 | 15.76 |
| Richest | 904 | 30.98 |

*Catholic or traditional or wakefata **SNNPR: South Nation, Nationalities, and People Region.*

### Obstetric-related characteristics of participants

Two-thirds, 1934 (66.3%) of the women had a history of births in the past five years and 2063 (70.7%) of the women had received ANC from a skilled provider. Among those who got ANC services, 1656 (56.8%) had four or more visits. One thousand two hundred and forty-five (43%) had their first ANC visit in the first three months of pregnancy. Of women who received ANC, 2635 (90.3%) had their blood pressure checked, 2412 (82.7%) had a blood sample taken, and 2292 (78.5%) had a urine sample taken "Table 2."

### Adequacy and consistency of antenatal care services in Ethiopia

Nationally provided antenatal care service was evaluated in terms of adequacy and consistency of ANC services based on the availability of information from mini-EDHS 2019. About 1656 (56.8%) women reported that they had at least four ANC visits from skilled attendants during their recent pregnancy, while only 1245 (43%) women-initiated ANC visits during the first trimester of gestation. Further, 1046 (35.8%) of the women received all six routine components of ANC services. Among those currently pregnant women, only 234(72.4%) received consistent antenatal care services "Table 3."

### Regional disparities of adequacy of ANC services in Ethiopia

The ANC services' adequacy was far less than 50%. It was higher in Addis Ababa, 88 (39.1%), and lowest in SNNPR, 20 (6.2%) and Gambela, 23 (8.6%) "Table 4."

### Regional disparities of provision of consistent ANC services in Ethiopia

The study evaluated the overall consistency of antenatal care (ANC) services, with 233 individuals receiving consistent ANC services and 90 individuals receiving inconsistent ANC services. The data indicates that 72.45% of the women received consistent ANC services "Fig 2."

The assessment of ANC service provision considered visit frequency, first initiation time, and completeness of services. Addis Ababa demonstrated the highest level of consistency in ANC service provision at 54.2%, followed by Dire Dawa at 45.5%. Benishangul and Harar had consistency rates of 34.55% and 34.3%, respectively. Conversely, Oromia (13.5%) and SNNPR (13.9%) exhibited poor ANC service consistency, ranking among the lowest in this aspect. It is evident that targeted efforts are needed to improve ANC service consistency in regions with lower rates "Table 5."

### Factors associated with adequacy of ANC services

Logistic regression analyses were conducted to identify factors linked to the adequacy of ANC services, using a 95% confidence interval and a significance level of $P < 0.05$. The final model indicated that education level, region, wealth, and literacy were key factors affecting the adequacy of ANC services in Ethiopia. Women with higher education were almost twice as likely to receive adequate ANC services compared to those without formal schooling [AOR = 1.77, 95% CI: (1.02, 3.09)]. Regional disparities were also significant: women from the Oromia region had a 51% lower likelihood (AOR = 0.49, 95% CI: 0.31, 0.79), from the Benishangul region a 59% lower likelihood (AOR = 0.41, 95% CI: 0.24, 0.69), from the SNNPR region a 70% lower likelihood (AOR = 0.30, 95% CI: 0.17, 0.53), and from the Gambela region a 71%

**Table 2. Obstetric-related characteristics of participants in Ethiopia, 2019E_DHS (n = 2918).**

| Variables | Frequency | Percent |
|---|---|---|
| Frequency of ANC visit | | |
| 1 | 141 | 4.8 |
| 2 | 353 | 12.1 |
| 3 | 768 | 26.3 |
| 4+ | 1656 | 56.8 |
| Received ANC from a skilled provider | | |
| Yes* | 2063 | 70.7 |
| No | 855 | 29.3 |
| Place of ANC follow-up | | |
| Government health facility | 2716 | 93.1 |
| Private hospital | 114 | 3.9 |
| NGO health facility | 88 | 3 |
| Components of ANC service | | |
| Blood pressure measured | 2635 | 90.3 |
| Blood sample taken | 2412 | 82.7 |
| Urine sample taken | 2292 | 78.5 |
| Given or bought iron tablets/ syrup | 2214 | 75.9 |
| Nutritional counseling | 2088 | 71.6 |
| Told about Signs of pregnancy complications or dangerous signs of pregnancy | 1648 | 56.5 |
| Births in the last five years | | |
| 1 | 1934 | 66.3 |
| 2 | 865 | 29.6 |
| 3 | 109 | 3.7 |
| 4 | 10 | 0.4 |
| Currently pregnant women | | |
| No or unsure | 2595 | 88.9 |
| Yes | 323 | 11.1 |
| Gestational Age at 1st ANC visit (n = 323) | | |
| ≤3 months | 66 | 20.4 |
| 4-6 months | 121 | 37.5 |
| 7 months | 44 | 13.6 |
| 8 + months | 92 | 28.5 |

Hint ANC: *Antenatal care; * Skilled providers include doctors, nurses, midwives, health officers, and health extension workers.*

lower likelihood (AOR = 0.29, 95% CI: 0.15, 0.54) of receiving adequate ANC services compared to women from the Tigray region. The wealth index was another significant factor. Women in the poorer, middle, richer, and richest wealth categories had greater odds of receiving adequate ANC services, with adjusted odds ratios of [AOR = 1.96, 95% CI: (1.15, 3.34)], [AOR = 2.28, 95% CI: (1.35, 3.87)], [AOR = 2.52, 95% CI: (1.49, 4.24)], and [AOR = 3.66, 95% CI: (2.10, 6.37)], respectively, when compared to women in the poorest wealth category "Table 6."

## Factors associated with the consistency of ANC services in Ethiopia

Women with higher levels of education were more likely to receive consistent antenatal care (ANC) compared to those with no education. Women attended secondary education [AOR: 11.02 (95% CI: 1.47–26.67) and higher education

**Table 3. Consistency of ANC Services in Ethiopia (n = 2918).**

| Variables | Frequency | Percent |
|---|---|---|
| Initiation time ANC visit during the recent pregnancy | | |
| Early | 1245 | 42.7 |
| Late | 1673 | 57.3 |
| Number of ANC visits | | |
| < 4 visits | 1262 | 43.2 |
| >=4 visits | 1656 | 56.8 |
| Components of ANC Services | | |
| Sufficient | 1046 | 35.8 |
| Insufficient | 1872 | 64.2 |
| Overall Consistency of ANC Services | | |
| Consistent ANC services | 234 | 72 |
| Inconsistent ANC services | 89 | 28 |

More than one-fifth (14.7%) of women were received adequate antenatal care (ANC) in Ethiopia "Fig 1."

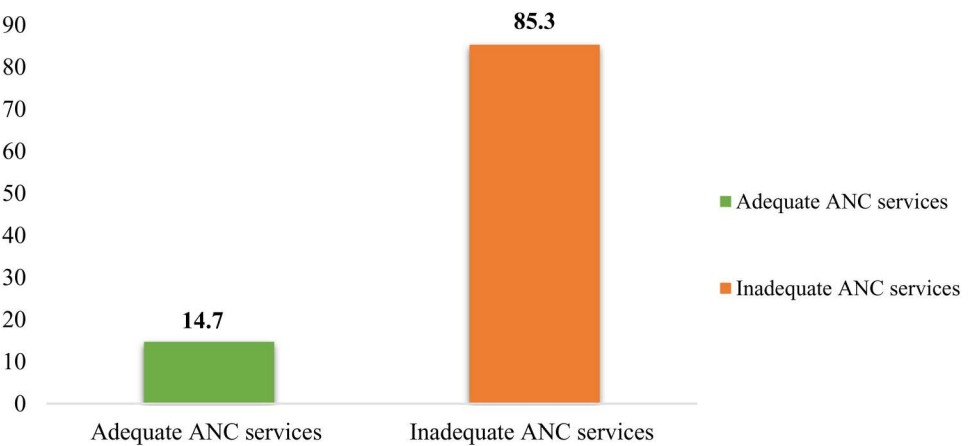

**Fig 1. Overall adequacy of antenatal care services among pregnant women in Ethiopia.**

[AOR: 15.73 (95% CI: 2.42–31.13) were 11 and 16 times more likely to received consistent ANC than women with no education respectively.

Women from Addis Ababa were 7.79 times more likely to receive consistent antenatal care (ANC) compared to women from Tigray [AOR = 2.79, 95% CI: (0.803, 9.658)]. Similarly, women from Dire Dawa were 2.67 times (AOR: 2.67 (95% CI: 0.85, 8.32) more likely to receive consistent ANC than women from Tigray "Table 7."

## Discussion

The study utilized 2019 Mini-EDHS data to assess the quality of prenatal care services in Ethiopia. The results showed that only 14.7% of women received adequate ANC services, indicating significant deficiencies in the national ANC delivery system. Interestingly, there was a notable difference in ANC service adequacy between the national findings and a prior study conducted in Southern Ethiopia, which reported a lower rate of 6.2%. This disparity could be attributed to the differing scopes of the studies, with the national study encompassing a larger geographic area while the Southern Ethiopia

**Table 4. Regional comparison of adequacy of ANC services in Ethiopia (n = 2918).**

| Variables | | Regional comparison of ANC adequacy | |
|---|---|---|---|
| | | Adequate (%) | Inadequate (%) |
| Region | Tigray | 71(22.1) | 251(77.9) |
| | Afar | 39(17) | 191(83) |
| | Amhara | 64(19.2) | 272(80.8) |
| | Oromia | 38(11) | 308(89) |
| | Somali | 8(9.3) | 78(90.7) |
| | Benishangul | 34(11.4) | 265(88.6) |
| | SNNPR | 20(6.2) | 306(93.8) |
| | Gambela | 23(8.6) | 245(91.4) |
| | Harari | 79(32.1) | 168(67.9) |
| | Addis Ababa | 88(39.1 | 139(60.9) |
| | Dire Dawa | 71(30.9) | 161(69.1) |

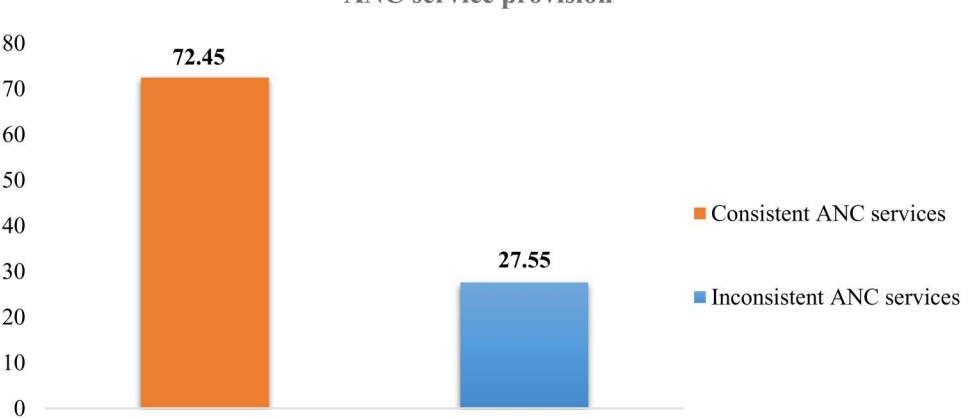

**Fig 2. Consistency of prenatal care services among pregnant women in Ethiopia.**

study focused specifically on the SNNP region [14,15]. Apart from the results that were previously mentioned, the study also showed that a considerable percentage of Ethiopian women did not make full use of prenatal care facilities. In particular, the findings showed that 66.3% of pregnant women did not have ANC during the first trimester, a critical time for early identification and treatment. In addition, 22.3% of the pregnant women had insufficient antenatal care, receiving less than four visits during their pregnancy [16].

The study discloses vital statistics regarding the initiation of antenatal care [17] and the number of ANC visits among women. The findings indicate that only 42.7% of women in Ethiopia initiated ANC during the first trimester, which is considered early and optimal for detecting and addressing potential pregnancy-related issues. Additionally, 43.2% of women had at least four ANC visits throughout pregnancy. Comparing these results to similar data from Nigeria, it is noteworthy that Ethiopia shows higher rates of ANC initiation in the first trimester. In Nigeria, only 18% of women started ANC during the first trimester [18], suggesting that Ethiopia has made relatively better progress in promoting early ANC engagement.

There are regional disparities in the proportion of women receiving adequate antenatal care (ANC) services in Ethiopia. Tigray, Harari, and Addis Ababa have higher rates, while SNNPR, Oromia, and Somalia report lower rates. These

**Table 5. Provision of consistent ANC service Regional Comparison of women of reproductive age in Ethiopia, 2024.**

| Variables | | Received consistent ANC service (n = 323) | |
|---|---|---|---|
| | | No (%) | Yes (%) |
| Region | Tigray | 27(75) | 9(25) |
| | Afar | 31(79.5) | 8(20.5) |
| | Amhara | 13(68.4) | 6(31.6) |
| | Oromia | 32(86.5) | 5(13.5) |
| | Somali | 10(76.9) | 3(23.1) |
| | Benishangul | 19(65.5) | 10(34.5) |
| | SNNPR | 31(86.1) | 5(13.9) |
| | Gambela | 19(82.6) | 4(17.4) |
| | Harari | 23(65.7) | 12(34.3) |
| | Addis Ababa | 11(45.8) | 13(54.2) |
| | Dire Dawa | 17(54.8) | 15(45.5) |
| Overall consistency | | 233(72.45) | 90(27.55) |

**Table 6. Multiple logistic regression analysis of the adequacy of ANC services in Ethiopian, 2019 E_DHS (n = 2918).**

| Variables | | Coef. | AOR (95% CI) | P-value |
|---|---|---|---|---|
| Educational status | No formal education | | 1 | |
| | Primary education | 0.02 | 1.02 (0.68, 1.52) | 0.94 |
| | Secondary education | 0.23 | 1.26 (0.74, 2.12) | 0.39 |
| | Higher education | 0.57 | 1.77 (1.02, 3.09) | 0.04* |
| Age group (years) | 15–24 years | | 1 | |
| | 25–34 years | 0.06 | 1.07 (0.80, 1.42) | 0.67 |
| | 35–49 years | 0.09 | 1.10 (0.71, 1.70) | 0.66 |
| Household wealth index | Poorest | | 1 | |
| | Poorer | 0.67 | 1.96 (1.15, 3.34) | 0.01* |
| | Middle | 0.83 | 2.28 (1.35, 3.87) | 0.01* |
| | Richer | 0.92 | 2.52 (1.49, 4.24) | 0.01* |
| | Richest | 1.30 | 3.66 (2.10, 6.37) | 0.01* |
| Residence | Urban | | 1 | |
| | Rural | −0.18 | 0.83 (0.58, 1.20) | 0.32 |
| Region | Tigray | | 1 | |
| | Afar | 0.01 | 1.01 (0.59, 1.73) | 0.98 |
| | Amhara | −0.17 | 0.85 (0.55, 1.30) | 0.45 |
| | Oromia | −0.71 | 0.49 (0.31, 0.79) | 0.01* |
| | Somali | −0.73 | 0.48 (0.18, 1.33) | 0.16 |
| | Benishangul | −0.90 | 0.41(0.24, 0.69) | 0.01* |
| | SNNPR | −1.20 | 0.30 (0.17, 0.53) | 0.01* |
| | Gambela | −1.25 | 0.29 (0.15, 0.54) | 0.01* |
| | Harari | −0.20 | 0.82 (0.53, 1.28) | 0.38 |
| | Addis Adaba | 0.02 | 1.02 (0.66, 1.59) | 0.93 |
| | Dire Dawa | −0.21 | 0.81 (0.52, 1.28) | 0.37 |
| Total children ever born** | $\beta = -0.010$ | | 0.99 (0.92, 1.07) | 0.79 |

Hint: 1= Reference group * = significant association at p-value <0.05 ** =Continuous variables.

**Table 7. Factors associated with the Consistency of ANC services in Ethiopia, 2019.**

| Variables Categories | | AOR | [95% CI] | | P-value |
|---|---|---|---|---|---|
| | No education | 1 | | | |
| Educational Attainment | Primary education | 1.27 | 0.54 | 2.99 | 0.58 |
| | Secondary education | 11.02 | 1.49 | 26.67 | 0.02* |
| | Higher education | 15.73 | 2.42 | 31.13 | 0.01* |
| Region | Tigray | 1 | | | |
| | Afar | 1.04 | 0.32 | 3.38 | 0.95 |
| | Amhara | 1.89 | 0.52 | 7.09 | 0.34 |
| | Oromia | 0.56 | 0.15 | 2.07 | 0.39 |
| | Somali | 1.02 | 0.19 | 5.24 | 0.99 |
| | Benishangul | 1.61 | 0.49 | 5.22 | 0.43 |
| | SNNPR | 0.49 | 0.13 | 1.86 | 0.30 |
| | Gambela | 0.40 | 0.08 | 2.07 | 0.27 |
| | Harari | 1.96 | 0.64 | 5.99 | 0.24 |
| | Addis Adaba | 2.79 | 0.80 | 9.66 | 0.04* |
| | Dire Dawa | 2.67 | 0.85 | 8.32 | 0.03* |

*Significant association at p-value <0.05.*

differences are likely due to variations in resources, infrastructure, and access to ANC services. Research indicates that factors such as socioeconomic status, healthcare facility accessibility, geographic location, and transportation significantly influence women's access to and use of ANC services [5,9,13,15,19]. Addressing these disparities requires targeted interventions to improve healthcare infrastructure, enhance the availability and accessibility of ANC services, and reduce socioeconomic barriers. Allocating resources and implementing strategies in regions with lower ANC adequacy rates will help improve maternal and child health outcomes across the country.

The study highlights the significance of educational status as a determinant of antenatal care service adequacy, with individuals having higher education demonstrating higher odds of receiving consistent ANC services. This underscores the importance of educational attainment in accessing and receiving adequate ANC. This result supports findings from Ethiopia [19], other sub-Saharan African research [20], a Study in Uganda [21], and a study in Nigeria [22]. The study also reveals a strong association between the household wealth index and ANC service adequacy, indicating that the odds of receiving consistent ANC services also increase as wealth increases. This result supports findings from [21,22]. This finding highlights the role of socioeconomic factors, particularly wealth, in accessing and utilizing ANC services effectively. Moreover, notable regional differences were observed, with individuals in certain regions, such as Oromia, Benishangul, SNNPR, and Gambela, having lower odds of receiving consistent ANC services than those in Tigray. These findings emphasize the need for targeted interventions and resource allocation to address regional disparities and ensure equitable access to ANC services across Ethiopia.

The finding of this study showed only 28% of women did not receive consistent prenatal care. Comparably, lower ANC coverage rates are problematic in several other Arabic nations. As an illustration, Morocco's coverage rate was 60.9%, whereas Mauritania's was 52.4%, and Saudi Arabia had a 79.9% ANC coverage rate in 2019 [17,23]. In contrast, some countries, such as Bahrain and the United Arab Emirates (UAE), stand out with exceptional ANC coverage, consistently nearing 100% [23]. The rates of women in Tigray and Afar who did not receive consistent ANC were 75% and 79.5%, respectively. Significant discrepancies were also seen in Oromia, Somali, SNNPR, and Gambela, where many women needed to receive ANC services consistently. Regional disparities in consistent ANC service provision result from differences in healthcare infrastructure, resource allocation, access to facilities, transportation challenges, socioeconomic

conditions, and cultural practices. Targeted interventions and investment in healthcare infrastructure are crucial in regions with lower service provision rates. Improving accessibility, raising awareness about ANC attendance, and addressing socioeconomic barriers can enhance access to ANC services. Addressing these disparities will promote equitable access to high-quality ANC services, improving Ethiopia's maternal and child health outcomes.

The study findings underscore the significance of educational attainment in the consistency of antenatal care services. Individuals with secondary education demonstrated significantly higher odds of receiving consistent ANC services than those without education, and individuals with higher education had even higher odds of ANC service consistency. This significant impact on educational level is similar to the study conducted in Saudi Arabia [17,23]. Regarding regional differences, individuals residing in Tigray had the highest likelihood of receiving consistent ANC services, while no statistically significant differences were observed in other regions compared to Tigray. These results emphasize the importance of educational attainment in ensuring ANC service consistency and suggest that regional disparities in Ethiopia have an insignificant impact on ANC service provision when compared to Tigray.

The implication of this study is that it discloses critical gaps in the adequacy and consistency of antenatal care services in Ethiopia, with noticeable disparities linked to region, education, literacy and wealth. Although many women are accessible to skilled ANC providers, few receive complete or adequate care services which indicate a quality shortfall in service delivery. Findings of this study indicated the need for focused interventions to advance Antenatal care (ANC) quality and it services, address regional and socioeconomic discrepancies and invest in women's literacy and education. Moreover, a move from emphasizing ANC coverage alone to enhancing service quality and equity is crucial for improving maternal and child health outcomes in Ethiopia.

### Limitations of the study

The study relied on self-reported data, which may be subject to recall bias. The study also needed to explore why ANC service use varies by region. Researchers need to conduct further research to understand the causes behind the variations.

### Conclusions

In conclusion, the study highlights the inadequate and inconsistent ANC services in Ethiopia. Regional disparities in ANC provision and factors such as education, wealth, and geographic location contribute to disparities in service utilization. It is crucial to promote ANC utilization, particularly among less educated and economically disadvantaged women, with a particular focus on rural areas. Improving accessibility and quality of ANC services in rural areas should be prioritized to ensure equitable access to care. Targeted interventions are needed to address the service gaps experienced by women with lower education levels, including effective communication strategies to address misconceptions and concerns. Moreover, addressing the significant regional differences in ANC consistency and adequacy requires implementing focused measures to bridge this gap. By improving ANC services, maternal and child health outcomes can be significantly enhanced throughout Ethiopia.

### Acknowledgments

The authors would like to thank the Demographic Health Survey program office for providing the needed data for this study and Jimma University for providing the internet.

### Author contributions

**Conceptualization:** Dessalegn Tamiru, Mohammed Areb, Gemechis Gunja, Million Abera, Desta Asefa, Negalign Berhanu, Yibeltal Siraneh, Fira Abamecha, Biru Abdissa Mizana.

**Formal analysis:** Dessalegn Tamiru, Biru Abdissa Mizana.

**Methodology:** Yibeltal Siraneh.

**Writing – original draft:** Mohammed Areb, Gemechis Gunja.

**Writing – review & editing:** Dessalegn Tamiru, Gurmesa Tura Debelew, Negalign Berhanu, Fira Abamecha.

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
