## [Decision Letter · Decision Letter 0]

21 Feb 2025

Dear Dr. Tamiru,

Thank you for submitting your manuscript to PLOS ONE. After careful consideration, we feel that it has merit but does not fully meet PLOS ONE’s publication criteria as it currently stands. Therefore, we invite you to submit a revised version of the manuscript that addresses the points raised during the review process.

We look forward to receiving your revised manuscript.

Kind regards,

Jennifer Tucker, PhD

Staff Editor

PLOS ONE

Journal Requirements:

2. Please note that your Data Availability Statement is currently missing the repository name and/or the DOI/accession number of each dataset OR a direct link to access each database. If your manuscript is accepted for publication, you will be asked to provide these details on a very short timeline. We therefore suggest that you provide this information now, though we will not hold up the peer review process if you are unable.

Reviewers' comments:

Reviewer's Responses to Questions

**Comments to the Author**

1. Is the manuscript technically sound, and do the data support the conclusions?

Reviewer #1: Partly

2. Has the statistical analysis been performed appropriately and rigorously?

Reviewer #1: Yes

3. Have the authors made all data underlying the findings in their manuscript fully available?

Reviewer #1: No

4. Is the manuscript presented in an intelligible fashion and written in standard English?

Reviewer #1: Yes

Reviewer #1: Consistency vs adequacy: Why these variables were included in consistency not in adequacy

why guest were considered eligible for survey

Please keep Study population consistent: is it last 1 year or 5 year

Tables: Please format table and keep decimal upto two places

Table 1: Please define household wealth index

Table 3: What is sufficient ANC services

Regional Disparities of Provision of consistent ANC services in Ethiopia: Please mention denominator and %

Factors associated with Adequacy of ANC services: Is literacy level different from education level?

Discussion: what is acceptable ANC?

rationale for regional disparities: Authors have mentioned, Socioeconomic status, healthcare facility accessibility, geographic location, and transportation can affect women's access to and utilization of ANC services. But some of these variables were not included in analysis

Limitation: The study did not explore the reasons behind the regional disparities in ANC adequacy.

**Do you want your identity to be public for this peer review?** For information about this choice, including consent withdrawal, please see our Privacy Policy

Reviewer #1: **Yes: ** Ankita Kankaria

---

## [Author Response · Author response to Decision Letter 1]

3 Mar 2025

Dear reviewer, we would like to express our sincere gratitude for your constructive comments and appreciation. We have carefully addressed each of your comment point by point.

We have also highlighted all the changes we made to the main manuscript.

1. Consistency vs adequacy

Response: Methods_Line 142- Dear reviewer we have operationalized these two variables. They are used to measure to consistency of ANC services and adequacy of ANC services. They are used to indicate different outcome variables.

1. Consistency of ANC visit: measured if the pregnant mother received services during antenatal care visits such as, identifying pre-existing health conditions, early detection of complications during pregnancy, health promotion and disease prevention, birth preparedness, and complication planning

2. Adequacy of antenatal care: The adequacy of antenatal care is measured using three indicators: early initiation of antenatal care (the first antenatal care visit made during the first trimester or three months), attending enough visits (at least four and above visits), and getting services at least once during the pregnancy care)

2. Why guest were considered eligible for survey?

Response: Methods_Line 123: We have addressed this comment, we paraphrased it.

‘‘The data were gathered through interviews with women aged 15 to 49 years, who were long-term residents of the selected households.’’

3. Please keep Study population consistent: is it last 1 year or 5 year

Response: Methods_Line 108: Thank you, we have addressed it.

The study population consisted of women who had at least one antenatal care visit during their most recent birth within the five years preceding the 2019 Ethiopian Mini-Demographic and Health Survey.

4. Tables: Please format table and keep decimal up to two places

Thank you, we have addressed it, we have corrected to two.

5. Table 1: Please define household wealth index

Response: Methods_Line 153: Now we have added the operational definition of Wealth Index:

The wealth index weights were derived using principal component analysis (PCA) based on data collected from household assets, utilities, and other variables from the Ethiopian Demographic and Health Survey. PCA was used to create latent factors that represent wealth, with the first factor serving as the household wealth score. These scores were then divided into five quintiles: lowest, second, middle, fourth, and highest.

6. Table 3: What is sufficient ANC services

Now we have added the operational definition of sufficient ante-natal care.

Methods_Line 146: Sufficient antenatal care: Sufficient antenatal care involves the recommended minimum of four timely visits, providing comprehensive services such as screenings, monitoring, and health counseling to ensure the well-being of both mother and baby.

7. Regional Disparities of Provision of consistent ANC services in Ethiopia: Please mention denominator and %

Response: Dear Reviewer, we have mentioned it in the table. A total of 323 individuals are included into the analysis. The number of individuals who are consistent and not consistent are mentioned in row (Table 5)

8. Factors associated with Adequacy of ANC services: Is literacy level different from education level?

Response: In this study educational status mean level of the education which the respondent attended.

9. Discussion: what is acceptable ANC?

Discussion Line_292: Sorry we have corrected it to adequate.

The study utilized 2019 Mini-EDHS data to assess the quality of prenatal care services in Ethiopia. The results showed that only 14.7% of women received adequate ANC services, indicating significant deficiencies in the national ANC delivery system.

10. Rationale for regional disparities: Authors have mentioned, Socioeconomic status, healthcare facility accessibility, geographic location, and transportation can affect women's access to and utilization of ANC services. But some of these variables were not included in analysis

Discussion line 313: Now, we have corrected it, here we want to show findings from other studies.

……Research indicates that factors such as socioeconomic status, healthcare facility accessibility, geographic location, and transportation significantly influence women's access to and use of ANC services….

11. Limitation: The study did not explore the reasons behind the regional disparities in ANC adequacy.

Response: Dear Reviewer, this study uses DHS data, a national survey conducted every five years. Some variables are selected at the national level to guide the survey. We, the authors, have no authority to add variables to further explore the reasons behind the regional disparities in ANC adequacy, as the data is collected by central statistics. We have also acknowledged this as a limitation in our report.

Line 364: Limitations of the study

The study relied on self-reported data, which may be subject to recall bias. The study also needed to explore why ANC service use varies by region. Researchers need to conduct further research to understand the causes behind the variations.

---

## [Decision Letter · Decision Letter 1]

9 May 2025

Sufficiency and Consistency of Antenatal Care Services in Ethiopia: Evidence from Mini Ethiopian Demographic and Health Survey

PONE-D-24-55077R1

Dear Dr. Tamiru,

We’re pleased to inform you that your manuscript has been judged scientifically suitable for publication and will be formally accepted for publication once it meets all outstanding technical requirements.

Kind regards,

Dereje Oljira Donacho, PhD

Academic Editor

PLOS ONE

Additional Editor Comments (optional):

Reviewers' comments:

Reviewer's Responses to Questions

**Comments to the Author**

Reviewer #2: All comments have been addressed

2. Is the manuscript technically sound, and do the data support the conclusions?

Reviewer #2: Yes

3. Has the statistical analysis been performed appropriately and rigorously?

Reviewer #2: Yes

4. Have the authors made all data underlying the findings in their manuscript fully available?

Reviewer #2: Yes

5. Is the manuscript presented in an intelligible fashion and written in standard English?

Reviewer #2: Yes

Reviewer #2: My Comments to authors

Thank you very much for giving me the opportunity to review this important research paper. In general, the paper is good and well written. Here below are my comments to improve the paper:

1. WHO updated their baseline recommendation to 8+ ANC contacts during one’s pregnancy. However, in this study, sufficient antenatal care and adequacy of antenatal care are operationalized based on four old ANC visits. What do you say about this issue? Can we say 4 visits are sufficient and adequate with other services?

2. What is the implication of this study? You have to show clear implication of this study in the discussion part

**Do you want your identity to be public for this peer review?** For information about this choice, including consent withdrawal, please see our Privacy Policy

Reviewer #2: No

---

## [Editor Report · Acceptance letter]

PONE-D-24-55077R1

PLOS ONE

Dear Dr. Tamiru,

I'm pleased to inform you that your manuscript has been deemed suitable for publication in PLOS ONE. Congratulations! Your manuscript is now being handed over to our production team.

Kind regards,

on behalf of

Dr. Dereje Oljira Donacho

Academic Editor

PLOS ONE